# Study of Angiogenic, Pro-Apoptotic, and Pro-Inflammatory Factors in Congenital and Acquired Cholesteatomas

**DOI:** 10.3390/jpm13081189

**Published:** 2023-07-26

**Authors:** Rolando Rolesi, Fabiola Paciello, Gaetano Paludetti, Eugenio De Corso, Bruno Sergi, Anna Rita Fetoni

**Affiliations:** 1Department of Otolaryngology Head and Neck Surgery, Università Cattolica del Sacro Cuore, 00168 Rome, Italy; gaetano.paludetti@policlinicogemelli.it (G.P.); eugenio.decorso@policlinicogemelli.it (E.D.C.); 2Fondazione Policlinico Universitario A. Gemelli IRCCS, 00168 Rome, Italy; fabiola.paciello@unicatt.it; 3Department of Neuroscience, Università Cattolica del Sacro Cuore, 00168 Rome, Italy; 4Department of Neuroscience, Reproductive Sciences and Dentistry-Audiology Section, University of Naples Federico II, 80131 Naples, Italy; annarita.fetoni@unina.it

**Keywords:** congenital cholesteatoma, acquired cholesteatoma, angiogenic factors, inflammation, cholesteatoma proliferation, bone erosion

## Abstract

Objectives. Despite recent advances in biomolecular research that have improved our knowledge of cholesteatoma pathogenesis, the reasons behind its highly variable clinical course are still not clarified. It has been proposed that biological signaling between peri-matrix and matrix cells could play a critical role in disease homeostasis. The aim of our study was to analyze the expression of inflammatory (IL-1β), hyper-proliferative (STAT-3, TGF-β), and angiogenic (VEGF-C, PDGFr) factors in congenital and acquired cholesteatomas (both in adults and children), which might correlate with the clinical features observed. We performed an experimental study on 37 patients (29 males and 8 females, ranging from 4 to 66 years of age) who were diagnosed with cholesteatoma between 2020 and 2021 in our institution. All patients underwent clinical, audiologic, and radiologic assessments. Bone erosion grading and staging of cholesteatoma growth were assessed through preoperative evaluation and intraoperative middle ear findings, according to the PTAM System proposed by the Japan Otological Society (2016). Retro-auricular skin specimens were intraoperatively collected in all patients. Skin and cholesteatoma samples were analyzed through histopathological, western blot, and immunohistochemical evaluations. The expression rate was measured to find out the differences between congenital and acquired cholesteatomas as well as between the adult and pediatric populations. Expression of angiogenic, inflammatory, and proliferative biomarkers is significantly increased in acquired cholesteatomas in children as compared to congenital and acquired forms in adults, in accordance with the higher stage of disease shown by imaging, surgical, and histological features. Our data suggest that pathways already supposed to be involved in the pathogenesis of cholesteatomas could be differently activated in more destructive forms, typically found in children. The identification of potential biomarkers of cholesteatoma aggressiveness could lead to more personalized management (timing of intervention, recurrence prevention) and the future identification of anti-growth/anti-proliferative agents as non-surgery therapeutic options.

## 1. Introduction

Cholesteatoma is characterized by abnormal keratinizing squamous epithelium growth inside the temporal bone, associated with local invasiveness that destroys multiple intra-temporal sub-structures as well as many intra- and extracranial complications [1,2]. Cholesteatoma usually appears as epithelial cell proliferation frequently surrounded by the formation of granulation tissue that is normally composed of three components: matrix, peri-matrix, and a pultaceous substance (including anucleate keratin squames, sebaceous, and necrotic tissues). Traditionally, cholesteatomas are classified into two different groups according to their different etiopathogenesis and clinical features: congenital cholesteatomas, specific to childhood, and acquired cholesteatomas, found in both children and adults. Congenital cholesteatoma frequently progresses without symptoms other than unilateral hearing loss and usually remains undetected until an incidental finding. In fact, it appears as a pearly white mass behind a normal tympanic membrane in patients without a history of otorrhea, previous otologic surgeries, or eardrum perforations. It represents approximately 3.7 to 24% of all presentations and 0.2–1.5% of all intracranial tumors [3]. Many theories have been posed to explain its congenital origin, mainly the occurrence of the ectodermal rests of the first and second branchial arches that residue into the middle ear [4].

As known, several hypotheses have been postulated for the origin of the primary acquired forms, which happen when there is recurrent middle ear inflammation and dysventilation that create a retraction pocket occurring in the *pars flaccida* region of the tympanic membrane, or as secondary acquired cholesteatomas usually caused by a pre-existing peripheral defect in the tympanic membrane that allows the epithelium to penetrate the middle ear [5,6,7,8,9,10]. The incidence of acquired cholesteatoma ranges between 9 and 13 cases and 3 to 15 cases per 100,000 subjects in adults and children, respectively. It is usually associated with chronic middle ear disease [1,11], such as chronic suppurative otitis media, characterized by a stable perforation of the tympanic membrane, recurrent otorrhea, otalgia, and hearing impairment. Interestingly, pediatric acquired cholesteatomas can be differentiated from their adult counterparts by their less favorable outcomes, as pediatric forms are more frequently infectious, more proliferative, and locally aggressive, with higher recurrence rates and an overall less favorable prognosis as compared to adult forms [10,11]. 

In the last two decades, advances in biomolecular research have improved our knowledge of cholesteatoma etiology and pathogenesis. Many studies to investigate the proliferative and bone-destroying activities of the cholesteatoma epithelium in both pediatric and adult patients were performed. However, conflicting results were often obtained, and many questions remain unanswered [12,13]. Histologically, cholesteatoma consists of three layers: a cystic content (keratin lamellae), a matrix composed of hyperproliferative stratified squamous epithelium, and a peri-matrix. The outermost layer is inflamed subepithelial connective tissue (granulation tissue) containing collagen fiber, fibrocytes, and inflammatory cells such as lymphocytes, histiocytes, plasma cells, and neutrophil leucocytes. Thus, it has been suggested that cholesteatoma may be a defective wound-healing process with persistent inflammatory and proliferative stages. On the other hand, immunohistochemically, it has been observed that granulation tissue of the peri-matrix plays a critical role in cholesteatoma homeostasis through the paracrine and autocrine interactions between peri-matrix fibroblasts and matrix keratinocytes [14]. Therefore, the aggressiveness of cholesteatoma might depend on an imbalance between peri-matrix and matrix factors, including a variety of angiogenic growth factors and pro-inflammatory cytokines. Thus, keratinocyte-derived cytokines activate paracrine secretion in the peri-matrix fibroblasts of other cytokines, including epidermal growth factor (EGF), tumor necrosis factor-alpha, platelet-derived growth factor (PDGF), and tissue growth factor TGF-a. In turn, these cytokines induce the differentiation, proliferation, and migration of matrix keratinocytes. In addition, autocrine loops are involved in maintaining tissue homeostasis; thus, some factors, such as TGF-a and TGF-b expressed in the hyperproliferative epithelium, regulate keratinocyte proliferation and differentiation [15,16,17,18]. 

Exploring alternative avenues through biomolecular research could be useful in the personalization of therapeutic choice-based phenotyping of cholesteatoma and likely lead to the development of nonsurgical options for the treatment of acquired cholesteatoma. To date, the purpose of our study has been to analyze the qualitative and quantitative expression patterns of inflammatory, hyper-proliferative, and angiogenetic factors in congenital and acquired cholesteatomas (in both adults and children), which might correlate with the clinical features observed and be used as potential biomarkers of outcome in patients who underwent surgery for cholesteatoma. 

## 2. Materials and Methods

### 2.1. Subject Selection

We performed an experimental study enrolling a group of patients who were diagnosed with cholesteatoma between 2020 and 2021 at our institution. Before treatments, all patients underwent clinical evaluation, temporal bone imaging (high-resolution computed tomography), and pure tone audiometry to confirm the cholesteatoma’s presence, morphological characteristics, and middle ear extension. As regards congenital forms, patients were considered eligible for our study in accordance with criteria proposed by Levenson et al.: diagnosis of a white mass behind an intact tympanic membrane, no previous invasive otologic procedures (e.g., trans-tympanic ventilation tube insertion, paracentesis), and no tympanic membrane perforation. All cases were confirmed by intraoperative findings. We excluded four adult recurrent cases and three pediatric patients with a recent episode of bilateral acute otitis media and tympanic membrane perforation. All included subjects had no significant complications at diagnosis. Overall, a total of 37 patients (29 male and 8 female), ranging in age from 4 to 66 years, were enrolled in our study. Cholesteatomas were classified into three groups as follows: group 1 (*n* = 10) congenital cholesteatoma; group 2 (*n* = 13) pediatric acquired cholesteatoma (age < 16 yo); and group 3 (*n* = 14) adult acquired cholesteatoma (age > 16 yo). Pathological features of the patients are listed in Table 1. 

For each patient, the staging of cholesteatoma growth was assessed through radiologic preoperative evaluation and intraoperative middle ear findings, according to the PTAM System proposed by the Japan Otological Society (2016) (Figure 1).

Bone erosion grading was evaluated by the detection of eroded bone spots according to the scoring system proposed by Mahmood et al. (2017) [19]. Retro-auricular skin specimens were intraoperatively collected in all patients. Skin and cholesteatoma samples were analyzed through histopathological, western blot, and immunohistochemical evaluations.

### 2.2. Immunofluorescence Analyses

The specimens were incubated with a blocking solution (1% BSA, 0.5% Tritonx-100, and 10% normal goat serum in PBS 0.1M), and then the slices were incubated overnight at 4 °C with a solution containing rabbit polyclonal anti-VEGF-C primary antibody (1:50, Santa Cruz Biotechnology, Inc., Dallas, TX, USA) or anti-IL-1ß (1:50, Santa Cruz Biotechnology, Inc.) or anti-PDGFR (1:100, Abcam, Cambridge, UK) or anti-TGF- ß (1:100, Bioss Antibodies, Woburn, MA, USA) or pSTAT3 (1:100, Cell Signaling, Beverly, MA, USA). All the primary antibodies used cross-reacted with human tissue. All specimens were incubated at room temperature for 2 h in labeled conjugated goat anti-rabbit secondary antibodies (AlexaFluor488 or 546, IgG, Invitrogen, Waltham, MA, USA) diluted 1:400 in 0.1 MPBS and DAPI stained (1:500 in 0.1 M PBS). Images (20 or 60×) were obtained with a confocal laser scanning system (NIKON TE, Confocal Head A1 MP, Tokyo, Japan) with an Ar/ArKr laser (for 488 nm excitation) and a HeNe laser (for 543 nm excitation). DAPI staining was imaged by two-photon excitation (740 nm, o140 fs, 90 MHz) performed with an ultrafast tuneable mode-lockedTi: sapphire laser (Chameleon, Coherent Inc., Santa Clara, CA, USA). 

### 2.3. Fluorescence Spectrum Intensity Analysis

To assess the expression of PDGRr, IL-1b, VEGFC, TGFb, and pSTAT3, we analyzed cryosections of cholesteatoma immunolabeled as described above. Each specimen was analyzed to recognize both matrix and peri-matrix structures. Multiple square regions of interest (ROIs) were defined to measure the immunofluorescence signal on the whole matrix and peri-matrix surfaces available on cryo-sections. The analysis was performed on five separate cryo-sections for each specimen, with an interval trimming of 10 μm. A planar spectrum intensity readout was performed in all immuno-labeled sections. Obtained data were processed using a specific-colored lookup table (spanning from black-blue to red-yellow) in order to better evaluate the immunofluorescence signal spectrum intensity of all analyzed samples. In all experimental groups, fluorescence planar readouts for matrix and peri-matrix tissues were mediated to obtain the optical density profile charts.

### 2.4. Western Blot

Tissues were collected on ice, stored at −80 °C, and homogenized using a RIPA buffer (Sigma-Aldrich, Milan, Italy). After centrifugation (12,000 rpm, 15 min, 4 °C), the supernatant’s aliquots were used to determine protein concentration using the Micro BCA kit (Pierce, Rockford, IL, USA). SDS-PAGE reducing sample buffer was added to the supernatant, and samples were heated to 95 °C for 5 min. Protein lysates (60 μg) were loaded onto 4–15% Tris-glycine polyacrylamide gels for electrophoretic separation. Colorburst^TM^ Electrophoresis markers (Sigma Aldrich, Milan, Italy) were used as molecular mass standards. Proteins were then transferred onto nitrocellulose membranes at 100 V for 2 h at 4 °C in a transfer buffer containing 25 mM Tris, 192 mM glycine, and 20% methanol. Membranes were stained with Ponceau S (Sigma), incubated for 1 h with blocking buffer (5% skim milk in Tris-buffered saline containing 0.5% Tween-20, TBST), and then incubated overnight at 4 °C with the following primary antibodies: VEGF-C primary antibody (1:100, Santa Cruz Biotechnology, Inc.); anti-IL-1ß (1:100, Santa Cruz Biotechnology, Inc.); anti-PDGFR (1:1000, Abcam); anti-TGF- ß (1:1000, Bioss Antibodies); pSTAT3 (1:1000, Cell Signaling). Membranes were then incubated with a horse-radish peroxidase-conjugated anti-rabbit (1:2000; Promega, Madison, WI, USA) IgG secondary antibody. The membranes were then washed, and the bands were visualized with an enhanced chemiluminescence detection kit (GE Healthcare, Buckinghamshire, UK). Protein expression was evaluated and documented using the UVItec Cambridge Alliance. 

### 2.5. Statistical Data

Data analyses of examination are presented as means ± standard errors of the mean (SEM), and differences were assessed using an ANOVA variance analyst (Statistica, Statsoft, Tulsa, OK, USA); a *p*-value < 0.05 was considered significant.

## 3. Results

### 3.1. Patients

According to data reported in Table 1 and Table 2, in our study, acquired pediatric cholesteatomas showed more aggressive and extensive growth as compared to congenital and adult acquired forms, in line with the literature [20]. 

### 3.2. Immunohistochemical Phenotyping of Cholesteatoma

Activation of angiogenic, inflammatory, and proliferative markers was evaluated by using immunohistochemistry and western blotting analyses in congenital, pediatric, and adult acquired cholesteatomas, distinguishing between matrix and peri-matrix areas. To evaluate the angiogenic pathways activated in both skin and cholesteatoma samples, immunostaining for VEGF-C and PDGFr was performed in all cholesteatoma specimens. A significantly higher PDGFr and VEGFC expression was detected as compared to the skin samples (compare Figure 2(A1) with Figure 2(B1–B3)). As shown in fluorescence spectrum intensity analysis on a pseudo-colored rainbow scale spanning from black-blue (low fluorescence intensity) to red-yellow (high fluorescence intensity), in all skin samples, VEGF-C and PDGFr expressions were faint, reaching the mean OD values of 5.07 ± 1.59 and 7.76 ± 2.45, respectively, as shown in a representative image in Figure 2. In congenital specimens, the mean VEGF-C optical density was 76.46 ± 4.30 at matrix level and 38.16 ± 1.88 in the peri-matrix (Figure 2(B1,C1)). In acquired pediatric cholesteatoma, mean VEGF-C OD values of 81.21 ± 1.95 and 190.98 ± 11.91 at matrix and peri-matrix levels, respectively (Figure 2(B2,C2)) were detected. Acquired adult cholesteatoma specimens showed a mean VEGF-C OD of 78.65 ± 3.36 (matrix) and a significantly lower peri-matrix value of 105.93 ± 4.11, as compared to acquired forms in children (Figure 2(B3,C3)). 

Similarly, PDGFr mean ODs were 63.24 ± 3.58 and 37.34 ± 1.64 in congenital cholesteatoma matrix and peri-matrix, respectively (Figure 2(D1,E1)), while a significantly higher PDGFr expression was found in acquired pediatric cholesteatomas, reaching a value of 101.46 ± 5.21 (matrix) and 227.55 ± 9.10 (peri matrix) (Figure 2(D2,E2)). In acquired adult forms, mean PDGFr OD values of 53.76 ± 1.75 and 94.97 ± 4.87 were detected at matrix and peri-matrix levels, respectively (Figure 2(D3,E3)). Taken together, the angiogenic biomarkers were more expressed in the peri-matrix of the acquired pediatric cholesteatoma as compared to the adults and less expressed in the congenital cholesteatoma. The immunostaining for TGF-β and pSTAT3 was performed to evaluate the proliferative activation of the cholesteatoma as compared to the skin samples. Both TGF-β and pSTAT3 expressions were faint in all skin preparations, reaching mean OD values of 4.69 ± 0.52, and 17.64 ± 1.05, respectively (Figure 3(A1,A2)). Congenital specimens showed an increased expression of TGF-β, reaching OD values of 45.09 ± 5.63 in the peri-matrix and 27.06 ± 1.32 at the matrix level, where the immunofluorescence signal appeared to be mainly located at the basal layer (Figure 3(B1,C1)). 

A much higher TGF-β expression was found in acquired cholesteatomas in the children’s population, where the mean TGF-β OD was about 79.08 ± 3.44 and 163.22 ± 9.40 in the matrix and peri-matrix, respectively (Figure 3(B2,C2)). Similarly, TGF-β expression increased in acquired adult cholesteatoma samples, although to a significantly lesser extent as compared to acquired forms in children, reaching mean OD values of 61.17 ± 6.31 and 94.45 ± 3.04 in matrix and peri-matrix, respectively (Figure 3(B3) and Table 3). 

As mentioned above, in all skin control samples, pSTAT3 labeling was faint (Figure 3(A2)). In all cholesteatoma specimens, a significant increase in pSTAT3 expression was mainly found at the matrix level. However, while a moderate increase in pSTAT3 expression was detected at the matrix layer of both congenital cholesteatomas (mean OD = 88.51 ± 5.97) and adult acquired forms (mean OD = 75.49 ± 2.72) (see Figure 3(D1,E1,D3,E3)), in children’s acquired specimens a significant further increase was detected (mean OD = 125.54 ± 4.88; *p* < 0.001) (Figure 3(D2,E2), and Table 3 and Table 4). Linear immunofluorescence spectrum readout showed a significantly higher pSTAT3 nuclear expression in children with acquired cholesteatoma as compared to other experimental groups (Figure 4H).

The involvement of the inflammatory processes was evaluated by using a pro-inflammatory cytokine, IL1-β immunostaining. In normal retro-auricular skin samples, the epidermis layer showed faint positive staining for IL1-β (mean OD = 17.35 ± 1.55) (Figure 4(A1)). In congenital cholesteatomas, matrix, and peri-matrix tissues were weakly stained for IL-1β (mean OD = 38.41 ± 1.04 and 39.12 ± 0.93 for matrix and peri-matrix, respectively) (Figure 4(B1,C1)). A significantly increased amount of IL-1β (*p* < 0.001) was detected in the peri-matrix tissue underlying the matrix of both adult and acquired pediatric cholesteatomas (OD = 108.44 ± 9.47 and 148.68 ± 4.07, respectively) (Figure 4(B2,C2,B3,C3,F)). 

Taken together, inflammatory and proliferative biomarkers are significantly more expressed in the peri-matrix of acquired cholesteatomas, especially in pediatric specimens. 

### 3.3. Western Blot Analyses

As shown in Figure 3L, western blot analysis confirmed immunofluorescence studies. In all analyzed samples, VEGFC and PDGFr, as well as pSTAT3, IL-1β, and TGF-β level expression, were basal in skin tissues with respect to cholesteatoma-related ones (Figure 4D–I). Consistent with immunofluorescence analysis, in acquired cholesteatomas, particularly in children, VEGF-C and PDGFr expression increased compared to congenital cholesteatoma samples (Figure 4D,E,I). Furthermore, the expression of the pro-inflammatory cytokine IL-1β was significantly higher in acquired cholesteatoma (Figure 4F,I). The phosphorylation of STAT3 and TGF- β expression showed the same trend, greatly increased in acquired pediatric forms as compared to other cholesteatomas (Figure 4G,H,I). Overall, immunostaining and quantitative analyses by using western blotting demonstrate that biomarkers for angiogenic, inflammatory, and proliferative biomarkers are significantly increased in acquired cholesteatomas and in children compared to adults, corresponding with the higher stage of disease shown by imaging, surgical, and histological features in the pediatric population as compared to adults or congenital cholesteatomas (Table 3 and Table 4). 

## 4. Discussion

As well-known so far, cholesteatoma represents a complex model of cell growth dysregulation in which more aggressive forms could follow the peri-matrix processes containing all factors involved in matrix growth as well as nearby bone structure resorption. The nature of its hyper-proliferative and osteolytic characteristics is still widely debated and seems to involve multiple factors, including genetic/epigenetic alterations to external stimuli, infections, inflammation processes, and activation of several signal transduction pathways. Despite recent advances in molecular biology improving our knowledge of the pathophysiology of cholesteatoma, its pathogenesis is a very complex process that may involve multiple mechanisms, and no single theory can completely explain its clinical characteristics such as aggressiveness, hyperproliferation, and recidivism [10].

Although the clinical profile of all cholesteatomas generally includes aggressive local growth associated with surrounding bone structure destruction, among Otologists it is well known that its clinical course can be highly variable, from static lesions causing little or no disability to rapidly invasive ones with severe complications [21]. For this reason, in the last few decades, some researchers have attempted to clarify the biological characteristics of cholesteatomas with highly aggressive clinical behavior. In this scenario, an exciting perspective regards the role of matrix/peri-matrix molecular interplay in the progression of cholesteatoma. It has been suggested that biological interactions between peri-matrix fibroblasts and matrix keratinocytes play a crucial role in cholesteatoma tissue homeostasis and growth. 

More specifically, several pro-inflammatory cytokines (IL-1α, IL-1β, IL-6, and IL-8) and other signaling proteins, such as parathyroid hormone-related protein (PTHrP), can be released by matrix keratinocytes. Many authors suggested these cytokines could induce peri-matrix fibroblast activation, leading to an increase in several other cytokines’ secretion, such as epidermal growth factor (EGF), tumor necrosis factor-alpha (TNFα), platelet-derived growth factor (PDGF), tumor growth factor alpha (TGFα), keratinocyte growth factor (KGF), and granulocyte macrophage colony stimulating factor (GM-CSF), all involved in the modulation of matrix inflammatory and proliferative/differentiation responses [10]. Besides this paracrine stimulation, these activated keratinocytes could increase local TGFα/β release, which contributes, via an autocrine loop mechanism, to matrix differentiation and proliferation [10,22,23]. Moreover, many of these cytokines promote peri-matrix angiogenesis (EGF, PDGF, IL-8, TGFα) and osteoclastogenesis/bone resorption (IL-1, IL-6, PTHrP, receptor activator of nuclear factor kappa-B ligand (RANKL), which are both responsible for cholesteatoma proliferation and local aggressiveness [10,24,25]. It is also supposed that Signal transducer and activator of transcription 3 (STAT3) could play a crucial role in cholesteatoma cell growth, as well as many other processes such as autophagy, cytoskeletal organization, and apoptosis inhibition [26]. 

Generally, pediatric cholesteatomas exhibit a more aggressive growth pattern and higher recurrence rates as compared with adults. As regards the different clinical features between congenital and acquired cholesteatomas in the pediatric population, it has been reported that hearing loss is present at clinical onset with the same frequency in both CC and AC. In contrast, the frequency of presentation of otorrhea, otalgia, and otorrhagia is considerably lower in CC. Although a higher rate of stapes superstructure damage is reported in CC in comparison with AC (probably due to different preferential cholesteatoma locations), recurrence rates and planned two-stage surgery are much more frequently required for AC as compared to CC [11]. 

For these reasons, the aim of our study was to establish a comparison of matrix/peri-matrix expression of angiogenic (VEGF-C, PDGFr), proliferative (pSTAT3, TGF-β), and inflammatory (IL-1β) factors to identify a simple biological profile possibly related to the clinical features in children and adult patients. According to our data, VEGF/PDGFr immunofluorescence signal, although overexpressed in all cholesteatomas, showed significantly higher values in acquired cholesteatomas as compared to congenital ones (*p* < 0.001), particularly at the peri-matrix level, where the significantly higher mean OD were detected in the children’s population (*p* < 0.001). These data support the hypothesis that increased vascularization of the cholesteatoma peri-matrix could act as a pivotal factor in its destructive feature. Similarly, IL-1 β—which is supposed to be another responsible for cholesteatomata’s bone erosion and local inflammation by promoting a leucocyte’s release of several proteolytic enzymes- was mostly overexpressed at the peri-matrix level of acquired cholesteatoma. This result is particularly expressed in younger patients, who exhibited extensive bone destruction as compared to congenital cholesteatoma, which is generally asymptomatic and associated with low bone injury. 

Moreover, our data showed that expression of TGF-β appeared to be slightly increased in the cholesteatoma matrix, mostly located at the basal layer, with a similar extent among congenital and acquired cholesteatomas. However, we found a noticeably increased TGF-β signal at the peri-matrix level of acquired forms in children (see Figure 2(B1–B3,C1–C3)) where immunofluorescence spectrum readouts revealed much higher values, up to 1.8 and 3.7-fold higher as compared to acquired forms in adults (*p* < 0.001) and congenital forms (*p* < 0.001), respectively. Taken together, our data on IL-1 and TGF-β expression in acquired cholesteatoma support the hypothesis that in more symptomatic cholesteatomas, activation of proinflammatory pathways could lead to increased TGF-β expression, which is one of the most relevant factors involved in cell proliferation and migration [18,27]. As regards pSTAT3 expression, previous studies demonstrated that pSTAT3 is mainly localized in both the nuclei and cytoplasm of basal and supra-basal cholesteatoma cells [26]. In our samples, pSTAT3 was over-expressed in all cholesteatoma specimens. However, since pSTAT3 represents the phosphorylated STAT3 that can translocate to the cell nucleus, thus acting as a transcriptional activator, we performed an analysis of the linear readout of immunofluorescence signal intensity on epithelium basal cells, which confirmed a much higher nuclear/cytoplasm ratio in children with acquired cholesteatoma as compared to congenital forms and acquired cholesteatoma in adults. Overall, our observations confirm that increased hyper-proliferative and angiogenic expression exists in all cholesteatoma epithelium, although our data compressively indicate an over-activation in child-acquired forms. In the past, other studies demonstrated that the main striking feature of cholesteatoma, namely its ability to reabsorb bone tissue adjacent to its peri-matrix, is associated with a higher local concentration of osteoclast progenitor cells as compared to normal adjacent bone tissue [15]. Our data agree with the hypothesis that in more locally invasive cholesteatomas, peri-matrix VEGF-C and PDGFr over-expression could participate in osteoclast local recruitment and/or differentiation, as VEGFs act as bone homeostasis regulators and exert paracrine effects on osteoclast differentiation processes. 

## 5. Conclusions

Even if our study has some limitations, such as the low number of biologic pathways investigated and the small sample size, our data may suggest that the angiogenic (VEGF, PDGFr) and pro-proliferative (TGF-b, STAT3) pathways already supposed to be involved in the pathogenesis of cholesteatomas could be differently modulated/activated in locally more destructive forms, typically found in children (Figure 5). Anyway, further studies will be needed to validate the current results since the precise identification of specific predictors of cholesteatoma aggressiveness could lead to more personalized management (timing of intervention, recurrence, prevention) and the future identification of anti-growth/anti-proliferative agents as non-surgery therapeutic options. 

## Figures and Tables

**Figure 1 jpm-13-01189-f001:**
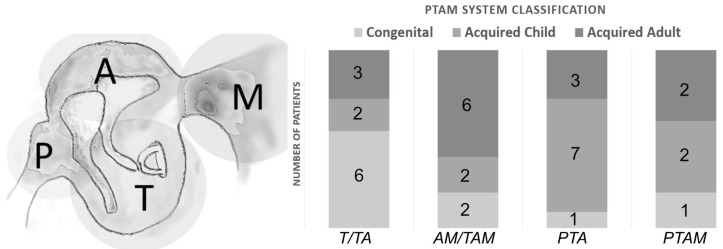
PTAM classification. We used the PTAM classification to evaluate the pathological extent of the cholesteatoma. In this system, the tympanomastoid cavity is divided into separate sections: (P) Protympanum, (T) Tympanic cavity, (A) Attic, and (M) Mastoid. The graph bars on the right show the different extents of cholesteatoma within PTAM sections in our experimental groups.

**Figure 2 jpm-13-01189-f002:**
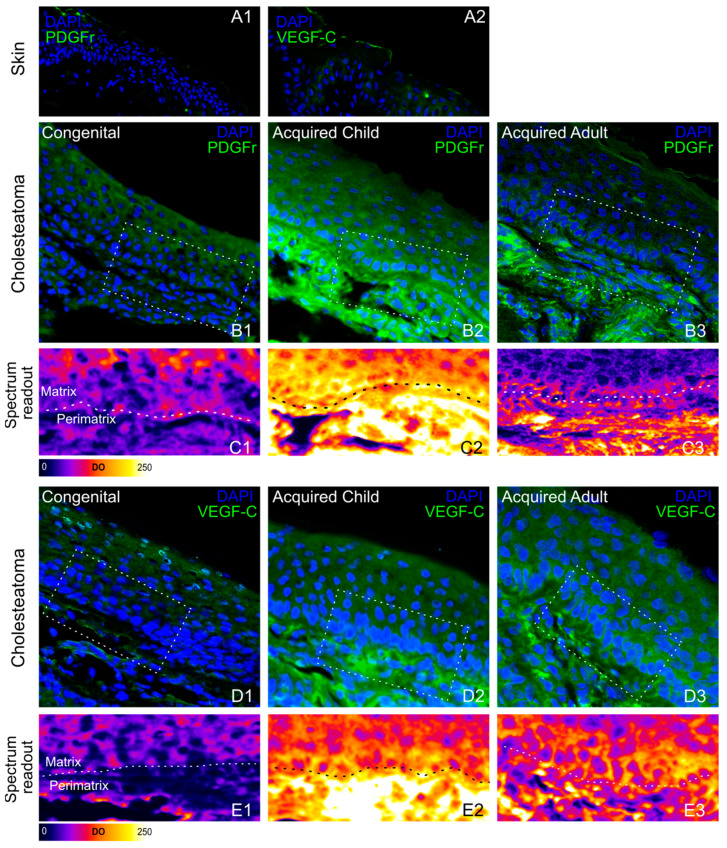
PDGFr and VEGF-C expression. Figure 2 shows representative images of skin (**A1**,**A2**) and cholesteatoma cryosections stained with DAPI (blue fluorescence) and Pdgfr (green fluorescence, (**B1**–**B3**)) or VEGF-C (green fluorescence, (**D1**–**D3**)). (**C1**–**C3**,**E1**–**E3**) show a specific colored lookup table to evaluate immunofluorescence signal intensity, spanning from black-blue (lower intensity) to red-yellow (higher intensity). Overall, a significant (*p* < 0.001) overexpression of both PDGFr and VEGF-C was detected in acquired pediatric cholesteatomas, particularly at the peri-matrix level.

**Figure 3 jpm-13-01189-f003:**
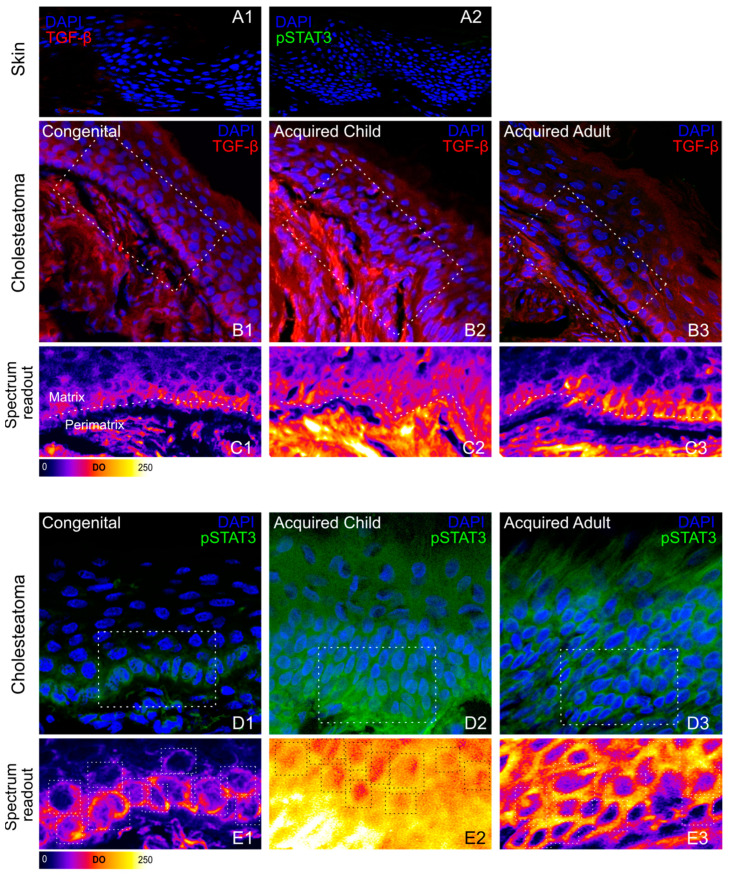
TGF-β and pSTAT3 expression. The figure shows representative images of skin (**A1**,**A2**) and cholesteatoma cryosections stained with DAPI (blue fluorescence), TGF-β (red fluorescence, (**B1**–**B3**)), or pSTAT3 (green fluorescence, (**D1**–**D3**)). (**C1**–**C3**,**E1**–**E3**) show a specific colored lookup table to evaluate immunofluorescence signal intensity, spanning from black-blue (lower intensity) to red-yellow (higher intensity). Both pro-proliferative factors are significantly overexpressed in pediatric acquired cholesteatoma (*p* < 0.001).

**Figure 4 jpm-13-01189-f004:**
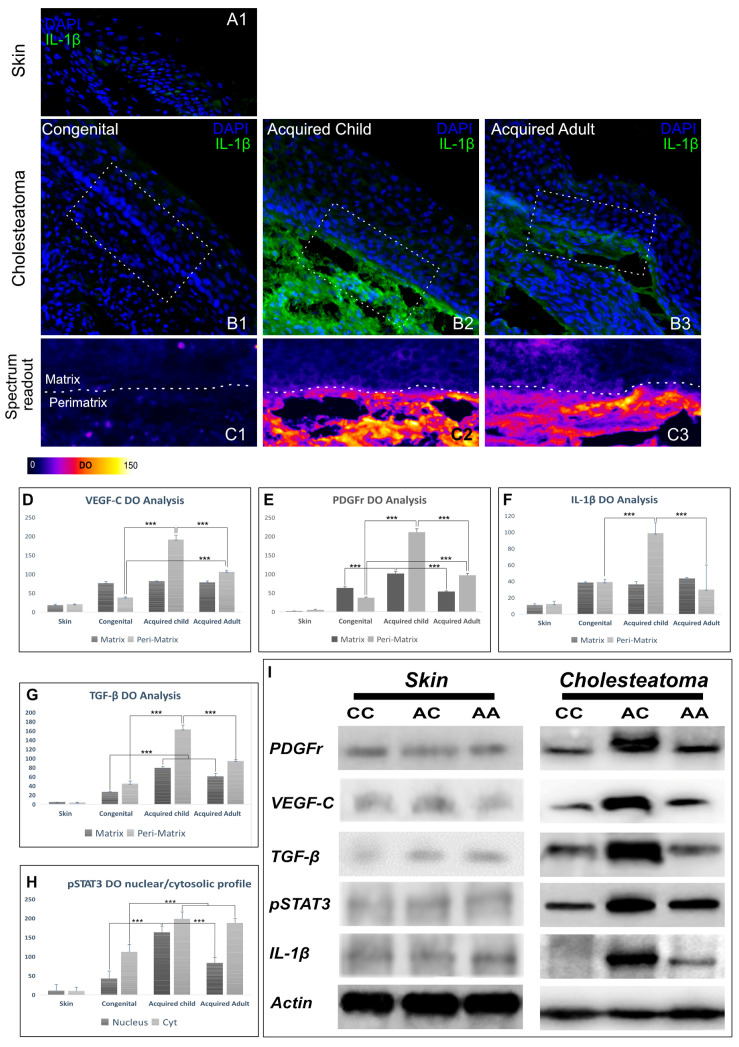
The figure shows representative images of skin (**A1**) and cholesteatoma cryosections stained with DAPI (blue fluorescence) and IL-1β (green fluorescence, (**B1**–**B3**)). (**C1**–**C3**) show a specific-colored lookup table to evaluate immunofluorescence signal intensity, spanning from black-blue (lower intensity) to red-yellow (higher intensity). Increased IL-1β expression was detected at the peri-matrix level of acquired cholesteatomas, particularly in pediatric patients. (**D**–**H**) immunofluorescence DO comparison among groups. (**I**) representative images of Skin and Cholesteatoma Western blot analysis. (CC) congenital cholesteatoma; (AC) pediatric acquired cholesteatoma; (AA) adult acquired cholesteatoma. Asterisks indicate significant differences among groups (*** *p* < 0.001).

**Figure 5 jpm-13-01189-f005:**
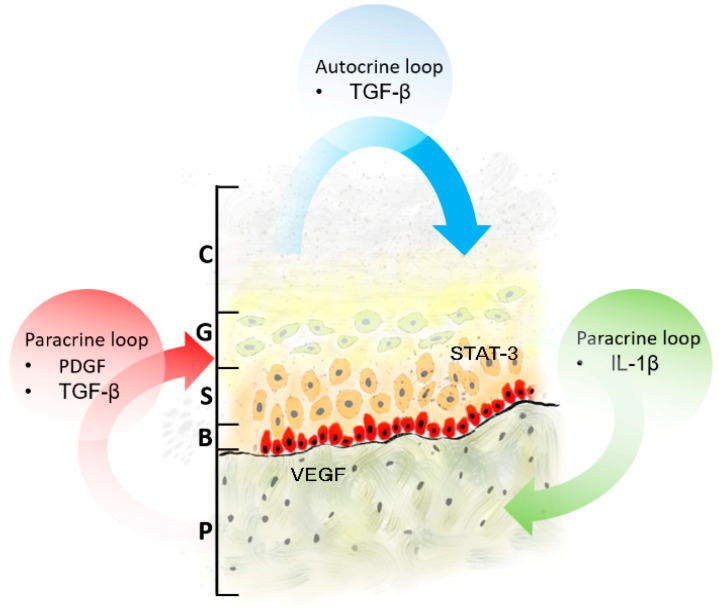
Molecular pathogenesis of cholesteatoma. Many Authors suggest that the interplay between matrix keratinocytes and peri-matrix fibroblasts is crucial for tissue homeostasis within cholesteatoma. Cholesteatoma differentiation and proliferation require both paracrine and autocrine signaling. (P) Peri-matrix; (B) basal layer; (S) stratum spinosum; (G) stratum granulosum; (C) stratum corneum.

**Table 1 jpm-13-01189-t001:** Patients features. In this table, we report the main symptoms (in percentage) affecting patients and their relative frequency. (CC) congenital cholesteatoma; (AC) Pediatric acquired cholesteatoma; (AA) adult acquired cholesteatoma.

	Intermitted Otorrhea	TM Perforation	Otalgia	Hearing Impairment
CC	10%	0%	40%	80%
CA	53.80%	69.20%	76.92%	53.84%
AA	42.86%	57.15%	42.85%	50%
	**Number of Patients**	**Congenital Form**	**Acquired Form (age < 16)**	**Acquired Form (age > 16)**
male	29	8	10	11
female	8	2	3	3
mean age		4.7	13.7	46.7

**Table 2 jpm-13-01189-t002:** According to the grading score proposed by Mahmood et al. (2017), cholesteatoma invasiveness was determined by counting the number of eroded bones and the presence of complications. Score 0–3: no or mild bone erosion (low invasive cholesteatomas); Score ≥ 4: severe bone erosion (Invasive cholesteatomas). (ICC) intracranial complication; (PCW) posterior canal wall. The bottom bars indicate the severity of bone erosion in our patients.

Bone	Grading Score	Bone	Grading Score
Scutum	Intact = 0, Eroded = 1	Sinus Plate	Intact = 0, Eroded = 1, Eroded with complications = 2
Auditory ossicles	Intact = 0, One ossicle eroded = 1, ≥2 ossicles eroded = 2	PCW	Intact = 0, Eroded without fistula = 1, Eroded with fistula = 2
Tegmen	Intact = 0, Eroded = 1, Eroded with ICC = 2	Mastoid	Free = 0, Eroded = 1, Abscess or fistula = 2
Facial Canal	Intact = 0, Dehiscent = 1, facial palsy = 2	Inner ear	Intact = 0, Erosion = 1, Fistula = 2
**Forms**	**Low Score (0–3)**	**Mid Score (3–6)**	**High Score (>6)**
Acquired adult	28.6%	50%	21.4%
Acquired child	7.7%	38.5%	53.8%
Congenital	70%	20%	10%

**Table 3 jpm-13-01189-t003:** Main immunofluorescence Optical Density (OD) values for skin, Congenital cholesteatoma (CC), Pediatric (AC), and Adult (AA) acquired cholesteatoma.

Optical Density (OD)	Skin		CC	AC	AA
VEGF-C	5.07 ± 1.59	Matrix Peri-matrix	76.46 ± 4.30 38.16 ± 1.88	81.21 ± 1.95 190.98 ± 11.91	78.65 ± 3.36 105.93 ± 4.11
PDGFr	7.76 ± 2.45	Matrix Peri-matrix	63.24 ± 3.58 37.34 ± 1.64	101.46 ± 5.21 227.51 ± 9.10	53.73 ± 1.75 94.97 ± 4.87
TGF- β	4.69 ± 0.52	Matrix Peri-matrix	27.06 ± 1.32 45.09 ± 5.63	79.08 ± 3.44 163.22 ± 9.40	61.17 ± 6.31 94.45 ± 3.04
pSTAT3	17.64 ± 1.05	Matrix Peri-matrix	88.51 ± 5.97 26.31 ± 4.19	125.54 ± 4.88 37.86 ± 10.50	75.49 ± 2.72 18.34 ± 7.05
IL-1β	17.35 ± 1.55	Matrix Peri-matrix	38.41 ± 1.04 39.12 ± 0.93	36.14 ± 11.46 148.68 ± 4.07	39.55 ± 4.30 108.44 ± 9.47

**Table 4 jpm-13-01189-t004:** Values of significance in the comparison among different groups.

MATRIX	Skin/CC	Skin/AC	Skin/AA	CC/AC	CC/AA	AC/AA
VEGF-C	*p* < 0.001	*p* < 0.001	*p* < 0.001	*p* = 0.31	*p* = 0.693	*p* = 0.474
PDGFr	*p* < 0.001	*p* < 0.001	*p* < 0.001	*p* < 0.001	*p* = 0.028	*p* < 0.029
TGF-β	*p* < 0.001	*p* < 0.001	*p* < 0.001	*p* < 0.001	*p* < 0.001	*p* = 0.022
pSTAT3	*p* < 0.001	*p* < 0.001	*p* < 0.001	*p* < 0.001	*p* = 0.063	*p* < 0.001
IL-1β	*p* < 0.001	*p* < 0.001	*p* < 0.001	*p* = 0.554	*p* = 0.518	*p* = 0.390
PERI-MATRIX	Skin/CC	Skin/AC	Skin/AA	CC/AC	CC/AA	AC/AA
VEGF-C	*p* < 0.001	*p* < 0.001	*p* < 0.001	*p* < 0.001	*p* < 0.001	*p* < 0.001
PDGFr	*p* < 0.001	*p* < 0.001	*p* < 0.001	*p* < 0.001	*p* < 0.001	*p* < 0.001
TGF-β	*p* < 0.001	*p* < 0.001	*p* < 0.001	*p* < 0.001	*p* < 0.001	*p* < 0.001
pSTAT3	*p* < 0.001	*p* = 0.071	*p* = 0.797	*p* = 0.289	*p* = 0.007	*p* = 0.085
IL-1β	*p* < 0.001	*p* < 0.001	*p* < 0.001	*p* < 0.001	*p* < 0.001	*p* < 0.001

## Data Availability

Not applicable.

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
