# Peer review of "Study of Angiogenic, Pro-Apoptotic, and Pro-Inflammatory Factors in Congenital and Acquired Cholesteatomas"

_jpm, 2023, doi:10.3390/jpm13081189_

Round 1
Reviewer 1 Report
Dear Authors
You need to review the English language of the manuscript. The errors were not flagrant but sometimes they interfered with text understanding, especially punctuation errors and word ordering of some sentences.
Now concerning the contents, first of all you need to make the PTAM System clear to the reader. Table 1 shows a cartoon of middle ear but does not explain the criteria employed by the system. Consequently the reader does not get a clear understanding of the stages in Table 1.
I do not consider your study "prospective experimental", as you did not evaluate an intervention - surgery, as it was part of usual healthcare procedures.
Table 2 also has some problems. I could not get a clear idea of the contribution of each clinical onset complication/symptom in your patient 3 groups. Perhaps showing percentages could help.
Another point, your patients were diagnosed according to Levenson's criteria; cholesteatoma stage was evaluated by PTAM System; bone erosion by Mahmood et al. 2017. You should represent the distribution of your cohort by a flow chart. For example, it is not clear to me the number of patients in your original cohort, and how many did not fulfill/passed Levenson's criteria.
You stated "ative CT findings shown in table 2 and 3, staging by using the preoperative CT findings matched surgical findings" - but where is this matching shown?
Your figures should have legends that summarize to the readers your findings, independently of the main text - you just placed short figure titles!
What statistical methods and analyses were performed to compare your results, especially the quantitative results, as fluorescence intensities?
You considered the skin samples as your controls for VEGF-C and PDGFr expressions, but also for TGF-β and pSTAT3 expressions. You informed results for skin OD values initially for VEGF-C and PDGFr - 5,07±1,59 and 7,76±2,45 respectively, and later for pSTAT3. But I missed those for TGF-β. It would be good to have all these OD values in a table with the results of statistical methods - p-values, for example, showing the differences among your 3 groups of patients and the controls (skin samples). You have a qualitative Table 4B, however you did not clarify the correspondence between OD values and number of plus signs. P-values are mentioned only in the Discussion section.
You did not show the western blot results of skin samples - your controls. You stated "data not shown". But you should show them, definitely.
Table 4A is a Figure that is more pertinent to the Discussion section. You could transfer it to the Discussion section, and moreover create a legend that clearly explains its contents and cite it in the text of the Discussion.
You stated: lines 323-330 "As regard to different clinical features between congenital and acquired cholesteatoma in children population, it has been reported that hearing loss is present at clinical onset, with the same frequency in both CC and AC. In contrast, the frequency of presentation of otorrhea, otalgia, and otorrhagia is considerably lower in CC, as well as mastoid pneumatization is more frequently good (??? seen?) in the CC Group, as compared to AC. Although a higher rate of stapes superstructure damage is reported in CC in comparison with AC (probably due to different preferential cholesteatoma location, recurrence rate and planned two-stage (and more than three times operation) are much more frequently required for AC as compared to CC [11]." However, you did not mention the incidence of the complications in your patients, and the differences between the 3 groups of your cohort.
You stated: lines 349-351: "However, we found a noticeably increase in TGF-β signal at peri matrix level of acquired forms in children, (see fig 2 B1-3, C1-3) were (??? where?) immunofluorescence spectrum readout revealed a much higher values (up to 1,8 and 3,7 fold higher as compared to acquired form in adults and congenital forms, respectively)." P-values are fundamental here.
You stated: lines 353-356: "Taken together, our data on IL-1 and TGF-β expression in acquired cholesteatoma, support the hypothesis that in more symptomatic cholesteatomas, proinflammatory pathways activation could lead to an increased TGF-β expression, which is one of the most relevant factors involved in cell proliferation and migration [18,27]." However, how did you avoid a recruitment bias of patients in your study? I mean you may have selected inadvertently the more severe cases of acquired cholesteatoma, and the milder cases of congenital cholesteatomas in children.
There are many problems concerning the English language use in the text, just few examples:
You stated: lines 323-330 "As regard to different clinical features between congenital and acquired cholesteatoma in children population, it has been reported that hearing loss is present at clinical onset, with the same frequency in both CC and AC. In contrast, the frequency of presentation of otorrhea, otalgia, and otorrhagia is considerably lower in CC, as well as mastoid pneumatization is more frequently good (??? seen?) in the CC Group, as compared to AC. Although a higher rate of stapes superstructure damage is reported in CC in comparison with AC (probably due to different preferential cholesteatoma location, recurrence rate and planned two-stage (and more than three times operation) [parenthesis inside parenthesis???] are much more frequently required for AC as compared to CC [11]."
You stated: lines 349-351: "However, we found a noticeably increase in TGF-β signal at peri matrix level of acquired forms in children, (see fig 2 B1-3, C1-3) were (??? where?) immunofluorescence spectrum readouts revealed a much higher values (up to 1,8 and 3,7 fold higher as compared to acquired form in adults and congenital forms, respectively)."
All over: angiogenetic; correct; angiogenic
Line 46: etio-pathogenesis; correct: etiopathogenesis
Line 58: pars flaccid - pars flaccida - English maintains the Latin name.
Lines 69-72: serious (lack of) punctuation problems
Line 75: peri matrix; correct: peri-matrix
From line 75 on, I gave up commenting the English language usage errors, as I think that the authors should revise all the stylistics, punctuation, and orthography of their text.
Author Response
Thank you for your revision.
Now concerning the contents, first of all you need to make the PTAM System clear to the reader. Table 1 shows a cartoon of middle ear but does not explain the criteria employed by the system. Consequently the reader does not get a clear understanding of the stages in Table 1.
Table 2 also has some problems. I could not get a clear idea of the contribution of each clinical onset complication/symptom in your patient 3 groups. Perhaps showing percentages could help.
As you can find in the revised version of the manuscript, we specified the contents of Table 1 and 2, according to your suggestions.
I do not consider your study "prospective experimental", as you did not evaluate an intervention - surgery, as it was part of usual healthcare procedures.
Another point, your patients were diagnosed according to Levenson's criteria; cholesteatoma stage was evaluated by PTAM System; bone erosion by Mahmood et al. 2017. You should represent the distribution of your cohort by a flow chart. For example, it is not clear to me the number of patients in your original cohort, and how many did not fulfill/passed Levenson's criteria
text was updated. We deleted the term prospective within the text. Moreover, as for the section “Materials and methods, we modified the paragraph about the “subject selection” following your suggestions (for more details see the modified text).
You stated "ative CT findings shown in table 2 and 3, staging by using the preoperative CT findings matched surgical findings" - but where is this matching shown?
About the “Results”, we corrected the text.
Your figures should have legends that summarize to the readers your findings, independently of the main text - you just placed short figure titles!
We added the figure legends at different level of the main text.
What statistical methods and analyses were performed to compare your results, especially the quantitative results, as fluorescence intensities?
Statistical section was updated in methods.
You considered the skin samples as your controls for VEGF-C and PDGFr expressions, but also for TGF-β and pSTAT3 expressions. You informed results for skin OD values initially for VEGF-C and PDGFr - 5,07±1,59 and 7,76±2,45 respectively, and later for pSTAT3. But I missed those for TGF-β. It would be good to have all these OD values in a table with the results of statistical methods - p-values, for example, showing the differences among your 3 groups of patients and the controls (skin samples). You have a qualitative Table 4B, however you did not clarify the correspondence between OD values and number of plus signs. P-values are mentioned only in the Discussion section.
According to your suggestion, we specified TGF-β skin OD values and added a table (table 4) which summarizes the main immunofluorescence OD values for all specimens. Moreover, we added a section within the table 4 that reports the values of significance in the comparison among different groups.
You did not show the western blot results of skin samples - your controls. You stated "data not shown". But you should show them, definitely.
Figure 3 was updated with western blot of skin samples
Table 4A is a Figure that is more pertinent to the Discussion section. You could transfer it to the Discussion section, and moreover create a legend that clearly explains its contents and cite it in the text of the Discussion.
Fig 4 was updated and replaced within the discussion section.
You stated: lines 323-330 "As regard to different clinical features between congenital and acquired cholesteatoma in children population, it has been reported that hearing loss is present at clinical onset, with the same frequency in both CC and AC. In contrast, the frequency of presentation of otorrhea, otalgia, and otorrhagia is considerably lower in CC, as well as mastoid pneumatization is more frequently good (??? seen?) in the CC Group, as compared to AC. Although a higher rate of stapes superstructure damage is reported in CC in comparison with AC (probably due to different preferential cholesteatoma location, recurrence rate and planned two-stage (and more than three times operation) are much more frequently required for AC as compared to CC [11]." However, you did not mention the incidence of the complications in your patients, and the differences between the 3 groups of your cohort.
Text was updated in the paraghraph “subject selection” (see All included subjects had no significant complication at diagnosis)
You stated: lines 349-351: "However, we found a noticeably increase in TGF-β signal at peri matrix level of acquired forms in children, (see fig 2 B1-3, C1-3) were (??? where?) immunofluorescence spectrum readout revealed a much higher values (up to 1,8 and 3,7 fold higher as compared to acquired form in adults and congenital forms, respectively)." P-values are fundamental here.
According to your suggestions, text was corrected and p-values added.
You stated: lines 353-356: "Taken together, our data on IL-1 and TGF-β expression in acquired cholesteatoma, support the hypothesis that in more symptomatic cholesteatomas, proinflammatory pathways activation could lead to an increased TGF-β expression, which is one of the most relevant factors involved in cell proliferation and migration [18,27]." However, how did you avoid a recruitment bias of patients in your study? I mean you may have selected inadvertently the more severe cases of acquired cholesteatoma, and the milder cases of congenital cholesteatomas in children.
We enrolled all patients who underwent specific surgical treatment for cholesteatoma, with the exception of recurrent cases and recent acute episode of otitis media. Certainly, our small simple of patients could be considered an important limitation of this study. However, we hope that our results could encourage further research.
There are many problems concerning the English language use in the text, just few examples:
You stated: lines 323-330 "As regard to different clinical features between congenital and acquired cholesteatoma in children population, it has been reported that hearing loss is present at clinical onset, with the same frequency in both CC and AC. In contrast, the frequency of presentation of otorrhea, otalgia, and otorrhagia is considerably lower in CC, as well as mastoid pneumatization is more frequently good (??? seen?) in the CC Group, as compared to AC. Although a higher rate of stapes superstructure damage is reported in CC in comparison with AC (probably due to different preferential cholesteatoma location, recurrence rate and planned two-stage (and more than three times operation) [parenthesis inside parenthesis???] are much more frequently required for AC as compared to CC [11]."
Text was corrected
You stated: lines 349-351: "However, we found a noticeably increase in TGF-β signal at peri matrix level of acquired forms in children, (see fig 2 B1-3, C1-3) were (??? where?) immunofluorescence spectrum readouts revealed a much higher values (up to 1,8 and 3,7 fold higher as compared to acquired form in adults and congenital forms, respectively)."
Text was corrected
All over: angiogenetic; correct; angiogenic
Line 46: etio-pathogenesis; correct: etiopathogenesis
Line 58: pars flaccid - pars flaccida - English maintains the Latin name.
Lines 69-72: serious (lack of) punctuation problems
Line 75: peri matrix; correct: peri-matrix
From line 75 on, I gave up commenting the English language usage errors, as I think that the authors should revise all the stylistics, punctuation, and orthography of their text.
text was revised
Reviewer 2 Report
The article well done. Worth publishing with some modifications:
a) Lines 56-68
on the pathogenesis of cholesteatoma it should be clarified that cholesteatoma is a complication of chronic suppurative otitis media which is characterized by a stable perforation of the tympanic membrane.
b) The whole article could be shorter
The article well done. Worth publishing with some modifications:
a) Lines 56-68
on the pathogenesis of cholesteatoma it should be clarified that cholesteatoma is a complication of chronic suppurative otitis media which is characterized by a stable perforation of the tympanic membrane.
b) The whole article could be shorter
Author Response
The article well done. Worth publishing with some modifications:
- a) Lines 56-68
on the pathogenesis of cholesteatoma it should be clarified that cholesteatoma is a complication of chronic suppurative otitis media which is characterized by a stable perforation of the tympanic membrane.
- b) The whole article could be shorter
Thank you for your revision. The text was modified according to your suggestion.
